# DEEP PROBABILISTIC VIDEO COMPRESSION

## ABSTRACT

We propose a variational inference approach to deep probabilistic video compression. Our model uses advances in variational autoencoders (VAEs) for sequential data and combines it with recent work on neural image compression. The approach jointly learns to transform the original video into a lower-dimensional representation as well as to entropy code this representation according to a temporally-conditioned probabilistic model. We split the latent space into local (per frame) and global (per segment) variables, and show that training the VAE to utilize both representations leads to an improved rate-distortion performance. Evaluation on small videos from public data sets with varying complexity and diversity show that our model yields competitive results when trained on generic video content. Extreme compression performance is achieved for videos with specialized content if the model is trained on similar videos.

## 1 INTRODUCTION

The transmission of video content is responsible for up to 80% of the consumer internet traffic, and both the overall internet traffic as well as the share of video data is expected to increase even further in the future (Cisco, 2017). Improving compression efficiency is more crucial than ever.

Today, a variety of video codecs exists that have reached an impressive performance. The most commonly used standard is H.264 (Wiegand et al., 2003), also known as Advanced Video Coding (AVC). More recent codecs include H.265 (Sullivan et al., 2012), also known as High Efficiency Video Coding (HEVC), and VP9 (Mukherjee et al., 2015). All of these existing codecs follow the same block based hybrid structure (Musmann et al., 1985) which essentially emerged from engineering out and refining this concept over decades. From a high level perspective, they differ in a huge number of smaller design choices and have grown to become more and more complex systems.

While there is room for improving the block based hybrid approach even further (Fraunhofer, 2018), the question is how much longer significant improvements can be obtained when following the same concept. Interestingly, in the context of image compression, deep learning approaches that are fundamentally different to existing codecs have already shown promising results (Ballé et al., 2018; 2016; Theis et al., 2017; Agustsson et al., 2017; Minnen et al., 2018), mostly within the past year.

Motivated by these successes for images, we propose a first step towards innovating beyond block-based hybrid codecs by framing video compression in a deep probabilistic context. To this end, we propose an unsupervised deep learning approach to encoding video. The approach simultaneously learns the optimal transformation of the video to a low-dimensional representation *and* a powerful predictive model that assigns probabilities to video segments, allowing us to efficiently entropy-code the discretized latent representation into a short code length.

Our end-to-end neural video compression scheme is based on sequential variational autoencoders (Bayer & Osendorfer, 2014; Chung et al., 2015; Li & Mandt, 2018) and the approach of Ballé et al. (2016) for discretizing and entropy coding a continuous latent representation. The transformations to and from the latent representation, known as the encoder and decoder, are parameterized by deep neural networks and are learned by unsupervised training on videos. We introduce both *local* latent variables, which are inferred from a single frame, and a *global* state, inferred from an entire segment, to efficiently store a video sequence. Furthermore, the trajectory of the latent variables is modeled stochastically by a deep probabilistic model. After training, the context-dependent predictive model is used to entropy code the latent variables into binary with an arithmetic coder.

| Ours (0.06 bpp) | H.265 (0.86 bpp) | VP9 (0.57 bpp) |
|---|---|---|

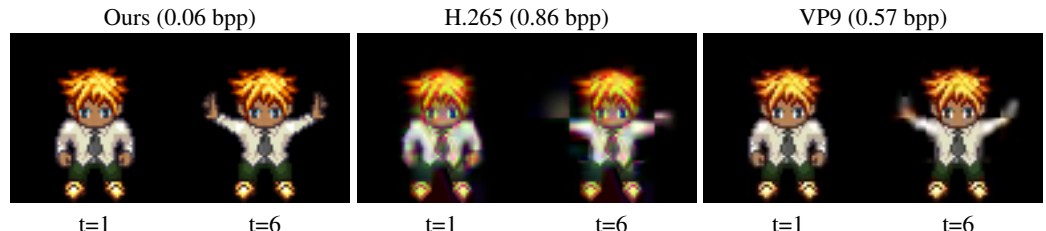

| t=1 | t=6 | t=1 | t=6 | t=1 | t=6 |
|---|---|---|---|---|---|

Figure 1: Reconstructed Sprites test video (bpp=0.06, PSNR=44.6 dB), H.265 (bpp=0.86, PSNR = 21.1 dB), and VP9 (bpp=0.57, PSNR = 26.0 dB), see Section 4. In contrast to our method, H.265 and VP9 show artifacts of block motion prediction. Our method uses a fraction of the bit rate.

As the first step towards a new approach, we focus on small resolution video ($64 \times 64$) and aim to efficiently capture temporal correlations. Figure 1 shows a test example of the possible performance improvements using our approach if the model is trained on similar content. The plots show two frames of a video, compressed and reconstructed by our approach and classical video codecs. One sees that fine granular details, such as the hands of the cartoon character, are lost in the classical approach due to artifacts from block motion estimation (low bitrate regime), whereas our deep learning approach successfully captures these details with less than 10% of the file length.

Our contributions can be understood as follows:

**1)** Deep probabilistic video compression. To the best of our knowledge, this is the first work to employ a variational autoencoder (VAE) in conjuction with discretization and entropy coding to build an end-to-end trainable video codec.

**2)** Global inference. Temporal redundancy in a video can be taken into account by a temporal prior on each frame or through an architectural design that encodes an entire video segment into a global state. We propose a model which incorporates both global and local latent variables and show that this produces a shorter code length for a given image quality.

**3)** Small bit rates. We perform experiments on three large public data sets of varying complexity and diversity. Performance is evaluated by rate-distortion curves. Our method is competitive with traditional codecs on small videos after training and testing on a diverse set of videos. Extreme compression performance can be achieved for videos with specialized content if the model is trained on similar videos.

**Paper Organization.** In Section 2, we summarize important related works before describing our method for deep video compression in Section 3. Section 4 discusses our experimental results, including quantitative results and a qualitative discussion.

## 2 RELATED WORK

The approaches related to our method fall into three categories: deep generative video models, neural image compression, and neural video compression.

**Deep Generative Video Models.** Several works have applied the variational autoencoder (VAE) (Kingma & Welling, 2014; Rezende et al., 2014) to stochastically model sequences (Bayer & Osendorfer, 2014; Chung et al., 2015). Babaeizadeh et al. (2018); Xu et al. (2018) use a VAE for stochastic video generation. He et al. (2018) and Denton & Fergus (2018) apply a long short term memory (LSTM) in conjunction with a sequential VAE to model the evolution of the latent space across many video frames. Li & Mandt (2018) separate latent variables of a sequential VAE into local and global variables in order to learn a disentangled representation for video generation. Vondrick et al. (2016) generate realistic videos by using a GAN (Goodfellow et al., 2014) to learn to separate foreground and background, and Lee et al. (2018) combine variational and adversarial methods to generate realistic videos. This paper also employs a deep generative model to model the sequential probability distribution of frames from a video source. In contrast to other work, our work learns a continuous latent representation that can be discretized with minimal information loss,

required for further compression. Furthermore, our objective is to convert the original video into a short binary description rather than to generate new videos.

**Neural Image Compression.** There has been significant work on applying deep learning to image compression. In Toderici et al. (2016; 2017), a LSTM based codec is used to capture spatial correlations of pixel values and can achieve different bit-rates without having to retrain the model. Ballé et al. (2016) perform image compression with a VAE and demonstrate how to approximately discretize the VAE latent space by introducing noise during training. This work is refined by (Ballé et al., 2018) which improves the prior model (used for entropy coding) beyond the mean-field approximation by transmitting side information in the form of a hierarchical model. Minnen et al. (2018) consider an autoregressive model to achieve a similar effect. These image codecs encode each image independently and therefore their probabilistic models are stationary with respect to time. In contrast, our method performs compression according to a non-stationary, time-dependent probability model which has much lower entropy per pixel.

**Neural Video Compression.** The use of deep neural networks for video compression is relatively new. Wu et al. (2018) perform video compression through image interpolation between reference frames using a predictive model based on a deep neural network. Chen et al. (2017) and Chen et al. (2018) use a deep neural architecture to predict the most likely frame with a modified form of block motion prediction and store residuals in a lossy representation. Since these works are based on block prediction, they are similar in function and in performance to existing codecs. Our method is not based on block motion estimation, and the full inferred probability distribution over the space of plausible subsequent frames (rather than residuals) is used for entropy coding.

## 3 DEEP PROBABILISTIC VIDEO COMPRESSION

The objective of lossy video compression can be defined as finding the shortest description of a video while tolerating a certain level of information loss. Classical video codecs try to predict intermediate frames based on block motion estimates, since information which can be predicted does not need to be stored. The residual error is then stored in a lossy transform representation. An end-to-end machine learning approach to encoding video, however, should simultaneously learn the appropriate predictive model and the optimal lossy transformation. This allows both to transform the video into a low dimensional latent representation and to then use the jointly learned predictive model to remove the remaining redundancy in the latents by entropy coding them to a short binary representation (Huffman, 1952; Langdon, 1984).

The challenge over image compression is that the data is now sequences of images. These exhibit strong temporal correlations in addition to the spatial correlations already present in images. A naive approach to neural video compression would be to encode the video frame-by-frame, using the marginal distribution of images as done in VAE image compression. This distribution does not capture the temporal correlations and therefore tends to have high entropy, leading to long code lengths. On the other hand, treating an entire video segment as an independent data point in the latent representation leads to data sparseness and poor generalization performance.

Therefore, we propose to use a temporally-conditioned prior distribution parameterized by a deep generative model to efficiently code the latent variables associated with each frame. By conditioning on nearby frames in the sequence, the predictive model can be more certain about the next frame, thus achieving a smaller entropy and code length. As detailed below, in addition to using a deep sequential probabilistic model, we propose an architecture that combines local and global information in the video. A global variable stores information that is common to the sequence of frames, while a local variable stores additional dynamical content.

In the following paragraphs, we describe our approach (see Fig. 3) in more detail. We describe the encoder and decoder models, the objective function, and the interplay between our deep probabilistic sequential model and entropy coding scheme.

**Decoder.** We propose a stochastic recurrent variational autoencoder to transform a sequence of frames $x_{1:T} = (x_1, \cdots, x_T)$ into a compressed representation of local latent variables $z_{1:T} = (z_1, \cdots, z_T)$, where each $z_i$ only depends on $x_i$. This model is refined to additionally include a global state $f$ similar to Li & Mandt (2018), resulting in the following probabilistic deep generative

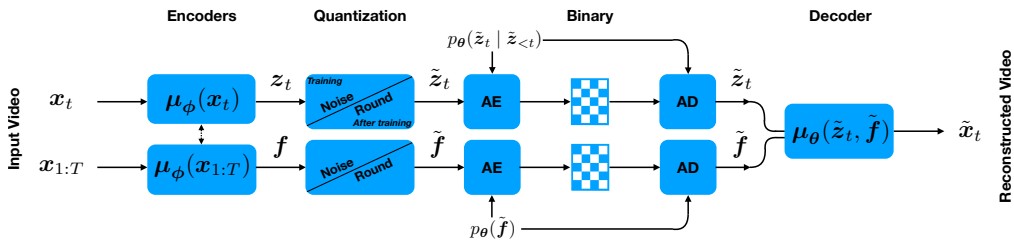

Figure 2: Operational diagram of our compression codec. A video segment is encoded into per-frame latent variables $z_t$ and per-segment global state $f$, which are then quantized and arithmetically encoded into binary according to the prior model. To recover an approximation to the original video, the latent variables are arithmetically decoded from the binary and passed through the decoder.

model:

$$p_{\theta}(\boldsymbol{x}_{1:T}, \boldsymbol{z}_{1:T}, \boldsymbol{f}) = p_{\theta}(\boldsymbol{f}) p_{\theta}(\boldsymbol{z}_{1:T}) \prod_{t=1}^{T} p_{\theta}(\boldsymbol{x}_t \mid \boldsymbol{z}_t, \boldsymbol{f}) \tag{1}$$

Above, $\theta$ is shorthand for the parameters. Each frame $\boldsymbol{x}_t$ at time $t$ depends on the corresponding latent variables $\boldsymbol{z}_t$ and global variables $\boldsymbol{f}$. The frame likelihood $p_{\theta}(\boldsymbol{x}_t|\boldsymbol{f}, \boldsymbol{z}_t)$ for reconstruction is the Laplace distribution, $\text{Laplace}\big(\boldsymbol{\mu}_{\theta}(\boldsymbol{z}_t, \boldsymbol{f}), \lambda^{-1}\mathbf{1}\big)$. The reason for its choice is that its log likelihood results in an $\ell_1$ loss, which typically produces sharper images than the $\ell_2$ loss for autoencoding images (Isola et al., 2016; Zhao et al., 2015). The decoder $\boldsymbol{\mu}_{\theta}(\cdot)$ is a function parameterized by neural networks. The prior distributions $p_{\theta}(\boldsymbol{f})$ and $p_{\theta}(\boldsymbol{z}_{1:T})$ will be discussed separately below.

After training, the reconstructed frame in image space is obtained from taking the most likely frame $\tilde{\boldsymbol{x}}_t = \arg\max p_{\theta}(\boldsymbol{x}_t|\boldsymbol{f}, \boldsymbol{z}_t) = \boldsymbol{\mu}_{\theta}(\boldsymbol{z}_t, \boldsymbol{f})$. Crucially, the decoder is conditioned both on global code $\boldsymbol{f}$ and time-local code $\boldsymbol{z}_t$.

**Encoder.** We employ amortized variational inference (Blei et al., 2017; Zhang et al., 2017; Marino et al., 2018) to predict a distribution over latent codes given the input video,

$$q_{\phi}(\boldsymbol{z}_{1:T}, \boldsymbol{f} \mid \boldsymbol{x}_{1:T}) = q_{\phi}(\boldsymbol{f} \mid \boldsymbol{x}_{1:T}) \prod_{t=1}^{T} q_{\phi}(\boldsymbol{z}_t \mid \boldsymbol{x}_t). \tag{2}$$

The global variables $\boldsymbol{f}$ are inferred from all video frames in a sequence and may thus contain static information, while $\boldsymbol{z}_t$ is only inferred from a single frame $\boldsymbol{x}_t$.

As will be explained in the paragraph on model-based entropy coding below, modifications to standard variational inference are required for compression. Instead of sampling from Gaussian distributions with learned variances, here we employ uniform distributions centered at their means:

$$\tilde{\boldsymbol{f}} \sim q_{\phi}(\boldsymbol{f} \mid \boldsymbol{x}_{1:T}) = \mathcal{U}\big(\hat{\boldsymbol{f}} - \tfrac{1}{2}, \hat{\boldsymbol{f}} + \tfrac{1}{2}\big); \quad \tilde{\boldsymbol{z}}_t \sim q_{\phi}(\boldsymbol{z}_t \mid \boldsymbol{x}_t) = \mathcal{U}\big(\hat{\boldsymbol{z}}_t - \tfrac{1}{2}, \hat{\boldsymbol{z}}_t + \tfrac{1}{2}\big). \tag{3}$$

The means are predicted by additional encoder neural networks $\hat{\boldsymbol{f}} = \boldsymbol{\mu}_{\phi}(\boldsymbol{x}_{1:T})$, $\hat{\boldsymbol{z}}_t = \boldsymbol{\mu}_{\phi}(\boldsymbol{x}_t)$ with parameters $\phi$. This corresponds to adding random noise $\epsilon_i \sim \mathcal{U}(-\tfrac{1}{2}, \tfrac{1}{2})$ to $\hat{\boldsymbol{f}}$ and $\hat{\boldsymbol{\mu}}$ in the inference process. The encoder and decoder neural networks are described in more detail in Appendix B.

**Prior Models.** The models that parameterize the learned prior distributions are ultimately used for entropy coding. Each dimension of the latent space has its own density model:

$$p_{\theta}(\boldsymbol{f}) = \prod_{i}^{\dim(\boldsymbol{f})} p_{\theta}(f^i) * \mathcal{U}(-\tfrac{1}{2}, \tfrac{1}{2}); \quad p_{\theta}(\boldsymbol{z}_{1:T}) = \prod_{t}^{T} \prod_{i}^{\dim(\boldsymbol{z})} p_{\theta}(z_t^i \mid \boldsymbol{c}_t) * \mathcal{U}(-\tfrac{1}{2}, \tfrac{1}{2}). \tag{4}$$

Above, indices refer to the dimension index of the latent variable and $\boldsymbol{c}_t$ is a time-dependent context. The convolution with uniform noise is to allow the priors to better match the true marginal distribution when working with the box-shaped approximate posterior in Eq. 3 (see Ballé et al. (2018) Appendix 6.2). This convolution has an analytic form in terms of the cumulative probability density.

The stationary density $p_{\boldsymbol{\theta}}(f^i)$ is parameterized by a flexible non-parametric, fully-factorized model from (Ballé et al., 2018). The density is defined by its cumulative and is built out of compositions of nonlinear probability densities, similar to the construction of a normalizing flow (Rezende & Mohamed, 2015).

Two dynamical models are considered to model the sequence $\boldsymbol{z}_{1:T}$. We propose a LSTM prior architecture which conditions on all previous frames in a segment: $p_{\boldsymbol{\theta}}(z_t^i \mid \boldsymbol{c}_t) \equiv p_{\boldsymbol{\theta}}(z_t^i \mid \boldsymbol{z}_{<t})$. We also considered a simpler model, which we compare against, with a single frame context: $p_{\boldsymbol{\theta}}(z_t^i \mid \boldsymbol{z}_{t-1})$ which is essentially a deep Kalman filter (Krishnan et al., 2015). A discussion of the connection between the amount of conditional context and the entropy is given in appendix A.

**Variational Objective.** The encoder (variational model) and decoder (generative model) can be learned jointly by minimizing the VAE loss function which consists of the KL divergence between the approximate and true posterior. In compression applications, however, one needs to adjust the trade-off between the bit rate and distortion. This is achieved by the $\beta$-VAE loss (Higgins et al., 2016; Mandt et al., 2016), which takes the form (up to constant terms)

$$- \mathbb{E}_{\tilde{\boldsymbol{f}}, \tilde{\boldsymbol{z}}_{1:T} \sim q}[\log p_{\boldsymbol{\theta}}(\boldsymbol{x}_{1:T} | \tilde{\boldsymbol{f}}, \tilde{\boldsymbol{z}}_{1:T})] - \beta \, \mathbb{E}_{\tilde{\boldsymbol{f}}, \tilde{\boldsymbol{z}}_{1:T} \sim q}[\log p_{\boldsymbol{\theta}}(\tilde{\boldsymbol{f}}, \tilde{\boldsymbol{z}}_{1:T})]$$

$$= \mathbb{E}_{\tilde{\boldsymbol{f}}, \tilde{\boldsymbol{z}}_{1:T} \sim q} \sum_{t=1}^{T} \|\tilde{\boldsymbol{x}}_t - \boldsymbol{x}_t\|_1 + \beta H \left[ q_{\boldsymbol{\phi}}(\tilde{\boldsymbol{z}}_{1:T}, \tilde{\boldsymbol{f}} \mid \boldsymbol{x}_{1:T}), p_{\boldsymbol{\theta}}(\tilde{\boldsymbol{f}}, \tilde{\boldsymbol{z}}_{1:T}) \right] \qquad (5)$$

where the reconstructed frame $\tilde{\boldsymbol{x}}_t = \boldsymbol{\mu}_{\boldsymbol{\theta}}(\tilde{\boldsymbol{z}}_t, \tilde{\boldsymbol{f}})$ and we have introduced a parameter $\beta$ to control the rate-distortion trade-off (Alemi et al., 2018). The Laplace parameter $\lambda$ was set to one.

The first term corresponds to the distortion and the second term is the cross entropy between the approximate posterior and the prior. The latter has the interpretation of the expected code length when using the prior distribution $p(\boldsymbol{f}, \boldsymbol{z}_{1:T})$ to entropy code the latent variables. This term is minimized for $p(\boldsymbol{f}, \boldsymbol{z}_{1:T}) = \mathbb{E}_{\boldsymbol{x}_{1:T}}[q(\boldsymbol{f}, \boldsymbol{z}_{1:T} | \boldsymbol{x}_{1:T})]$, that is, when the empirical distribution of codes matches the prior model. For our choice of generative model, the cross entropy separates into two terms $H\left[ q_{\boldsymbol{\phi}}(\boldsymbol{f} | \boldsymbol{x}_{1:T}), p_{\boldsymbol{\theta}}(\boldsymbol{f}) \right]$ and $H\left[ q_{\boldsymbol{\phi}}(\boldsymbol{z}_{1:T} | \boldsymbol{x}_{1:T}), p_{\boldsymbol{\theta}}(\boldsymbol{z}_{1:T}) \right]$.

**Model-based Entropy Coding.** Sequential VAEs can be used to reduce the dimensionality of a video by performing a lossy transformation. However, the reduction in dimensionality does not mean the video is optimally compressed since there still exists redundancy in the form of temporal correlations in the sequence of the latent space variables. Sequential VAEs may include a temporal prior to model this redundancy, but to actually remove this redundancy and achieve a compact binary representation, the latent space has to be entropy coded. This is the distinguishing element between variational autoencoders and compression algorithms.

Crucially, the entropy coder needs a discrete vocabulary, which is obtained by rounding the latent state ($\hat{\boldsymbol{f}}$ and $\hat{\boldsymbol{z}}_{1:T}$) after training. Care must be taken such that the quantization is approximated in a differentiable way during training. Ballé et al. (2016) address the problem of discretization of a continuous latent space in the context of VAE image compression by introducing noise in the inference process. By adding uniform noise to the most-likely inferred latent variables, the VAE is prevented from storing information in the latent space on length scales smaller than the discretization bin size. As such, rounding (after training) does not significantly affect the image reconstruction.

Our VAE framework exactly leads to such injection of noise with width one; it corresponds to our choice of a box-shaped approximate posterior distribution, centered at the maximally-likely values for the latent variables. Besides dealing with quantization, for efficient entropy coding we also need an estimate of the frequency of atoms in order to obtain short file sizes. We can obtain this from our prior model.

## 4 EXPERIMENTS

In this section, we present the experimental results of our work. We first describe the video datasets, performance metrics, and competing methods in Section 4.1. This is followed by a quantitative analysis of our performance in terms of rate-distortion curves in Section 4.2 and then finally qualitative results in Section 4.3. We report that our method can achieve extreme compression ratios on videos with specialized content if it is trained on similar videos and that the performance is

content-dependent. Our method is also comparable with modern codecs when trained on videos with diverse content. We find that the inclusion of the global state is more efficient than using solely local variables. Implementation details of our proposed models are provided in Appendix B.

## 4.1 DATASETS, METRICS, AND METHODS

In this work, we train separately on three video datasets of increasing complexity with frame size $64 \times 64$. **1) Sprites.** The simplest dataset consists of videos of Sprites characters from an open-source video game project, which is used in (Reed et al., 2015; Mathieu et al., 2016; Li & Mandt, 2018). The videos are generated from a script that samples the character action, skin color, clothing, and eyes from a collection of choices and have an inherently low-dimensional description (*i.e.* the script that generated it). **2) BAIR.** BAIR action-free robot pushing dataset (Ebert et al., 2017) consists of a robot pushing objects on a table, which is also used in (Babaeizadeh et al., 2018; Denton & Fergus, 2018; Lee et al., 2018). The video is more realistic and less sparse, but the content is specialized since all scenes contain the same background and robot, and the depicted action is simple since the motion is described by a limited set of commands sent to the robot. The first two datasets are uncompressed and no preprocessing is performed. **3) Kinetics600.** The last dataset is the Kinetics600 dataset (Kay et al., 2017) which is a diverse set of YouTube videos depicting human actions. The dataset is cropped and downsampled, which removes compression artifacts, to $64 \times 64$.

**Metrics.** We evaluate our method based on bit rate in bits per pixel (bpp), and distortion measured in average frame peak signal-to-noise ratio (PSNR), which is related to the frame mean square error. In the appendix, we also report on multi-scale structural similarity (MS-SSIM) (Wang et al., 2004) which is a perception-based metric that approximates the perceived change in structural information.

**Comparisons.** We wish to study the performance of our proposed local-global architecture with LSTM prior (LSTMP-LG) by comparing to other approaches. To study the effectiveness of predictive model for entropy coding, we introduce our baseline model LSTMP-L which has only local states with LSTM prior $p_{\boldsymbol{\theta}}(\boldsymbol{z}_t \mid \boldsymbol{z}_{<t})$. To study the efficiency of incorporating global and local states, we shows our baseline model KFP-LG which has both global and local states but with a weak predictive model $p_{\boldsymbol{\theta}}(\boldsymbol{z}_t \mid \boldsymbol{z}_{t-1})$, a deep Kalman filter (Krishnan et al., 2015). We also provide the performance of H.264, H.265, and VP9 codecs. Traditional codecs are not optimized for small resolution videos. However, their performance is far superior to neural or classical image compression methods (applied to compress video frame by frame), so their performance is presented for comparison. Codec performance is evaluated using the open source FFMPEG implementation in constant rate mode and distortion is varied by adjusting the constant rate factor. Unless otherwise stated, performance is tested on videos with 4:4:4 chroma sampling and on test videos with $T = 10$ frames.

## 4.2 QUANTITATIVE ANALYSIS: RATE-DISTORTION TRADE-OFF

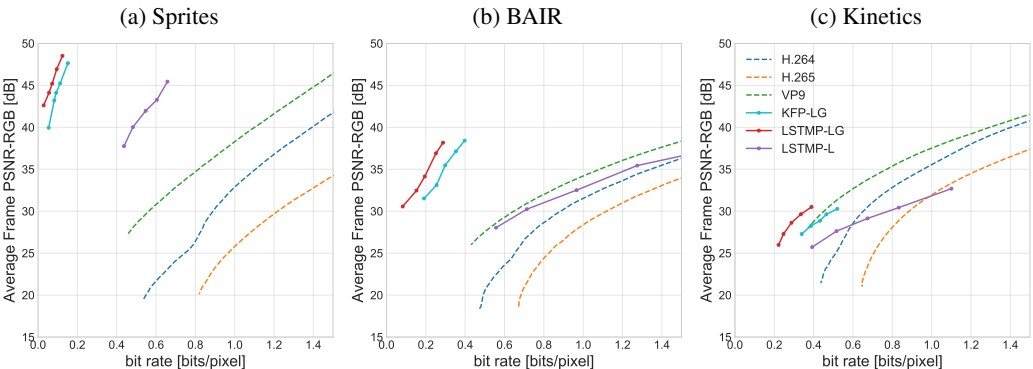

Figure 3: Rate-distortion curves on three datasets measured in PSNR (higher corresponds to lower distortion). Legend shared. Solid lines correspond to our models, with LSTMP-LG proposed.

Quantitative compression performance is evaluated in terms of rate-distortion theory. This characterizes the trade-off between the binary representation size and average distortion. For a fixed image quality setting, a video codec produces an average bit rate on a given dataset. By varying the image

quality setting, a curve is traced out in the rate-distortion plane. The curves for our method are generated by varying $\beta$ (Equation 5).

The rate-distortion curves for our method, trained on three datasets and measured in PSNR, are shown in Fig. 3. Higher curves indicate better performance. From the Sprites and BAIR results, one sees that our method has the ability to drastically outperform traditional codecs when focusing on specialized content. By training on videos with a fixed content, the model is able to learn an efficient representation for such content and the learned priors capture the empirical data distribution well (Appendix C). The results from training on the more diverse Kinetics videos also outperform or are competitive with standard codecs and better demonstrate the performance of our method on general content videos. Results measured in MS-SSIM (Appendix D) show similar behavior.

The first observation is that the LSTM prior outperforms the deep Kalman filter prior in all cases. This is because the LSTM model has a longer memory, allowing the predictive model to be more certain about the trajectory of the local latent variables. This, in turn, results in shorter code lengths.

**Global Variables.** The VAE encoder has the option to store information in local or global variables. The local variables are modeled by a temporal prior and can be efficiently stored in binary if the sequence $z_{1:T}$ can be sequentially predicted with relative certainty from the context. The global variables, on the other hand, provide an architectural approach to removing temporal redundancy since the entire segment is stored in one global state without temporal structure. We find that the local-global architecture (LSTMP-LG) outperforms the local architecture (LSTMP-L) on all datasets, demonstrating the usefulness of a hybrid approach which partially encodes the entire video segment in a global state along with extra frame-by-frame information stored as a sequence.

During training, the VAE learns to utilize the global and local information in the optimal way. The utilization of each variable can be visualized by plotting the average code length of each latent state, which is shown in Fig. 4. The VAE learns to significantly utilize the global variables even though $\dim(z)$ is sufficiently large to store the entire content of each individual frame. This provides further evidence that it is more efficient to incorporate global inference over several frames. The entropy in the local variables initially tends to decrease as a function of time, which supports the benefits from our predictive models. Note that our approach relies on sequential decoding, prohibiting a bi-directional LSTM for the local state.

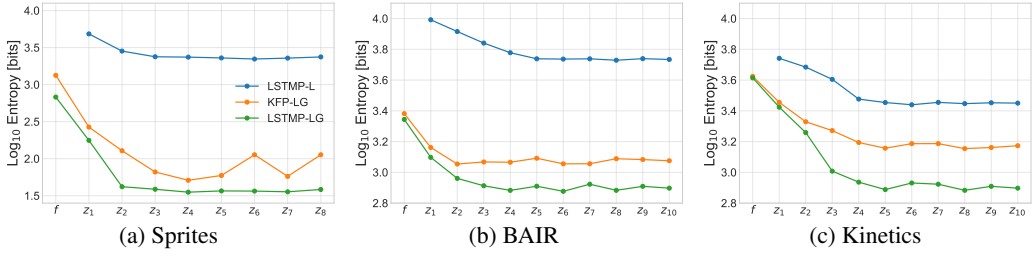

Figure 4: Average bits of information stored in $f$ and $z_{1:T}$ with PSNR 43.2, 37.1, 30.3 for different models in (a, b, c). Entropy drops with the frame index as the models adapt to the video sequence.

### 4.3 QUALITATIVE RESULTS

Now we discuss the qualitative performance of our method. We have shown that a deep neural approach to encode video (LSTMP-LG architecture) can achieve competitive results with traditional codecs with respect to PSNR or MS-SSIM (Appendix D) metrics overall on low-resolution videos. Test videos from the Sprites and BAIR datasets after compression with our method are shown in Fig. 1 and Fig. 5 (left), respectively, and compared to modern codec performance. Our method achieves a superior image quality at a significantly lower bit rate than H.264/H.265 and VP9 on these specialized content datasets. This is perhaps expected since traditional codecs cannot learn efficient representations for specialized content. Furthermore, fine-grained motion is not accurately predicted with block motion estimation. The artifacts from our method are more clearly displayed in Fig. 5 (right). Our method tends to produce blurry video in the low bit-rate regime but does not suffer from the block artifacts present in the H.265/VP9 compressed video.

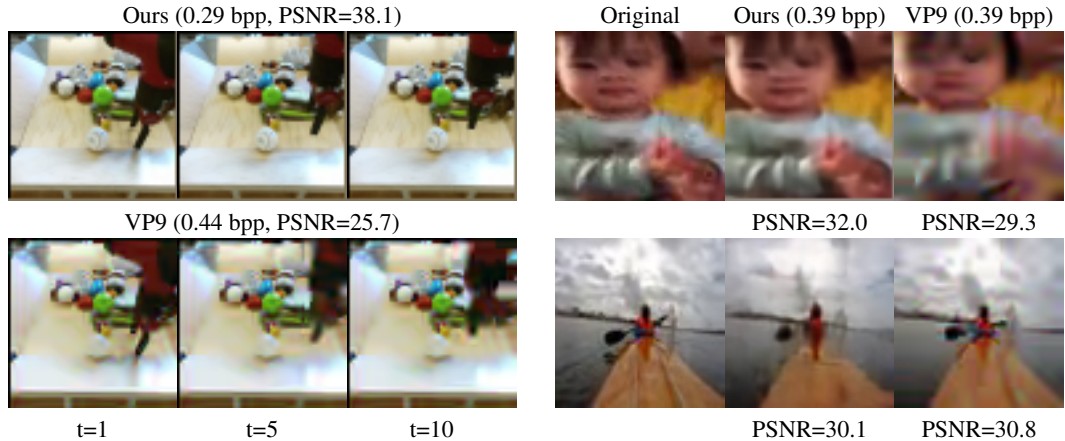

Figure 5: Compressed videos by our LSTMP-LG model and VP9 in the low bit rate regime. Our approach achieves better quality on specialized content (BAIR, left) and comparable visual quality on generic video content (Kinetics, right) compared to VP9.

**Limitations.** When training on specialized content videos, our method tends to fit to the type of content, leading to artifacts which can be more abstract and perhaps undesirable for a general purpose codec. For example, at a low bit-rate setting the codec tends to leave out new kinds of objects which were not present in the training data, or to misinterpret content. This is a big departure from the behavior of traditional codecs. Such undesirable defects can be avoided, however, by training on a general content training set or keeping a larger local latent state.

Our current paper focused on small scale videos. One future avenue is to extrapolate our method to full resolution videos, where the dimension of the latent representation must scale with the resolution of the video in order to achieve good reconstruction preformance. Currently, the GPU memory limits the maximum size of the latent dimension for the local/global architecture due to the presence of fully-connected layers to infer global and local states. This effectively limits the maximum achievable image quality of our method as well as restricts our method to short, low-resolution input video. While we showed that this architecture was very efficient for small videos in the strongly compressed regime, a future architecture may focus more on convolutional structures and avoidance of fully connected networks.

## 5    CONCLUSIONS

We have proposed a deep probabilistic modeling approach to video compression. Our method simultaneously learns to transform the original video into a lower-dimensional representation as well as the temporally-conditioned probabilistic model for entropy coding. The best performing proposed architecture splits up the latent code into global and local variables and yields competitive results on low resolution videos. For video sources with specialized content, deep probabilistic video coding allows for a significant increase coding performance. This could be interesting for transmitting specialized content such as teleconferencing or sports broadcasting.

We have shown viability on low resolution videos, and in future work, it is interesting to see how our LSTMP-L will work with convolutional LSTM as the predictive model in full-resolution videos and longer video sequences. Furthermore, there is much potential in investigating how additional "side information" could aid the predictive model. Thus we think that our work is a first step into a new direction for video coding which opens up several exciting avenues for future work.

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

## A    ENTROPY CODING

Predictive modeling is important in the entropy coding stage. Theoretical compression ratios can be computed from the cross entropy between the true probability distribution (measured empirically) and a model for the probabilities. A better model that more accurately captures the true certainty about the next symbol can achieve a lower entropy and thus a smaller bit rate. Entropy coding is a solved problem since there exist coding schemes, *e.g.* arithmetic coding, which approximately achieve the the entropy, up to extra information that is sent through the stream for practical reasons, and this bit rate is optimal for long messages (Shannon, 2001). Modeling, however, is difficult and leaves much room for improvement.

In this work, after the latent representation has been quantized to $N$ finite symbols, the variables are losslessly compressed into binary. The global latent variables $\boldsymbol{f}$ are static and are entropy coded according to a stationary distribution. The optimal lossless compression algorithm encodes the $i$-th symbol $f^i$ with a number of bits set by $\log \tilde{p}(f^i)$ on average, where $\tilde{p}$ is the true probability for $f^i$. The theoretical code length in bits after entropy coding the global variables with the prior distribution $p_{\boldsymbol{\theta}}(\boldsymbol{f})$ is computed from the cross entropy:

$$H\big[q_{\boldsymbol{\phi}}(\boldsymbol{f}|\boldsymbol{x}_{1:T}), p_{\boldsymbol{\theta}}(\boldsymbol{f})\big] = -\mathbb{E}_{\boldsymbol{f}\sim q} \sum_{i=1}^{\dim(\boldsymbol{f})} \log_2 p_{\boldsymbol{\theta}}(f^i), \tag{6}$$

where index $i$ refers to the dimension index of latent variable $\boldsymbol{f}$.

The cross entropy is minimized on average in the limit that $p_{\boldsymbol{\theta}}(\boldsymbol{f}) = \mathbb{E}_{\boldsymbol{x}_{1:T}}[q_{\boldsymbol{\phi}}(\boldsymbol{f}|\boldsymbol{x}_{1:T})] \equiv \tilde{p}(\boldsymbol{f})$, in which case the cross entropy becomes the Shannon entropy. In this work, we quote theoretical compression rates by computing the cross entropy between the variational distribution and the prior. However, we have explicitly checked that this performance is achieved by an arithmetic coder implementation which uses the probabilities from our learned prior.

For the case of the sequence $\boldsymbol{z}_{1:T}$, the theoretical description length of the sequence is given by the following cross entropy:

$$H\big[q_{\boldsymbol{\phi}}(\boldsymbol{z}_{1:T}|\boldsymbol{x}_{1:T}), p_{\boldsymbol{\theta}}(\boldsymbol{z}_{1:T})\big] = -\mathbb{E}_{\boldsymbol{z}_{1:T}\sim q} \sum_{t=1}^{T} \sum_{i=1}^{\dim(\boldsymbol{z})} \log_2 p_{\boldsymbol{\theta}}(z_t^i \mid \boldsymbol{c}_t) \tag{7}$$

where $\boldsymbol{c}_t$ is the context which may in principle depend on the previous appearing symbols in the sequence, the future symbols, or the global variables.

Consider the true probability distribution $\tilde{p}(\boldsymbol{z}_t|\neg\boldsymbol{z}_t)$, conditioned on all other $\boldsymbol{z}_t$ in the sequence both forward and backward in time. Averaging over all future elements of the sequence leads to a simplified distribution which only captures the conditional distribution based on a backwards in time context $\tilde{p}(\boldsymbol{z}_t|\boldsymbol{z}_{<t})$. One can average over long-distance context to arrive at an even simpler condition disribution $\tilde{p}(\boldsymbol{z}_t|\boldsymbol{z}_{t-1})$, or average out all context to obtain the marginal distribution $\tilde{p}(\boldsymbol{z}_t)$. We are guaranteed that the Shannon entropies are progressively larger since

$$H\left[\tilde{p}(\boldsymbol{z}_t)_{t=1:T}\right] \geq H\left[\tilde{p}(\boldsymbol{z}_t|\boldsymbol{z}_{t-1})_{t=1:T}\right] \geq H\left[\tilde{p}(\boldsymbol{z}_t|\boldsymbol{z}_{<t})_{t=1:T}\right] \geq H\left[\tilde{p}(\boldsymbol{z}_t|\neg\boldsymbol{z}_t)_{t=1:T}\right]. \tag{8}$$

Equality holds when the mutual information between $\boldsymbol{z}_{1:T}$ and the additional context vanishes.[1] The left-most entropy corresponds to treating each frame independently as in image compression. By improving the amount of context in the probabilistic model, the entropy and code length can be potentially reduced. We illustrate this point in the experiment section by demonstrating that the LSTM prior model, which has a longer contextual scope, produces a shorter code length than the deep Kalman filter prior.

## B    MODEL ARCHITECTURE

The specific implementation details of our model are now described. We describe the two baseline models, LSTMP-L and KFP-LG, and the best-performing LSTMP-LG model.

---

[1]This is equivalent to the fact from statistical physics that entropy of a system cannot decrease when averaging over a subset of the degrees of freedom.

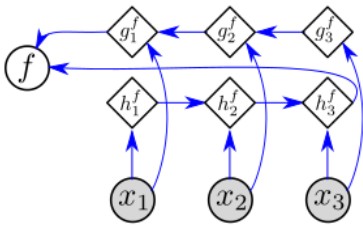

Figure 6: Inference network diagram for the global state $\boldsymbol{f}$. The features from the video segment are processed by a bi-directional LSTM (with hidden states $\boldsymbol{g^f}$, $\boldsymbol{h^f}$) which is used to infer the global state.

**LSTMP-L.** Our proposed baseline model LSTMP-L, which is introduced to study the efficiency of the global state for capturing temporal redundancy, contains only local latent variables $\boldsymbol{z}_t$ (the global state $\boldsymbol{f}$ is omitted). The local state for each frame $\boldsymbol{z}_t$ is inferred from each frame $\boldsymbol{x}_t$. LSTMP-L employs the same encoder and decoder architectures from Ballé et al. (2016). The encoder $\boldsymbol{\mu_\phi}(\boldsymbol{x}_t)$ infers each $\boldsymbol{z}_t$ independently by a five-layer convolutional network. For layer $\ell = 1$, the stride is 4, while a stride of 2 is used for layer $\ell = 2, 3, 4, 5$. The padding is 1 and the kernel size is $4 \times 4$ for all layers. The number of filters used for the Sprites video, for $\ell = 1, 2, 3, 4, 5$, are 192, 256, 512, 512 and 1024, respectively. For the more realistic video (BAIR and Kinetics video), the number of filters used at layer $\ell = 1, 2, 3, 4, 5$ are 192, 256, 512, 1024 and 2048, respectively. The decoder $\boldsymbol{\mu_\theta}(\boldsymbol{z}_t)$ is symmetrical to the encoder $\boldsymbol{\mu_\phi}(\boldsymbol{x}_t)$. With this architecture, the dimension of the latent state $\boldsymbol{z}_t$ is 1024 for Sprites and 2048 for BAIR and Kinetics video. The prior for the latent state corresponding to the first frame, $p_{\boldsymbol{\theta}}(\boldsymbol{z}_1)$, is parametrized by the same density model defined on Appendix 6.1 of Ballé et al. (2018). The conditional prior $p_{\boldsymbol{\theta}}(\boldsymbol{z}_t \mid \boldsymbol{z}_{<t})$ is parameterized by a normal distribution convolved with uniform noise (see Eq. 4). The means and (diagonal) covariance of the normal distribution are predicted by an LSTM with hidden state dimension equal to the dimension of the latent state $\boldsymbol{z}_t$.

**LSTMP-LG.** LSTMP-LG is our proposed model in this paper which uses an efficient latent representation by splitting latent states into both global states and local states as well as the use of an effective LSTM predictive model for entropy coding. Now we describe the inference network. The two encoders $\boldsymbol{\mu_\phi}(\boldsymbol{x}_{1:T})$ and $\boldsymbol{\mu_\phi}(\boldsymbol{x}_t)$ begin with a convolutional architecture to extract feature information. The global state $\boldsymbol{f}$ is inferred from all frames by processing the output of the convolutional layers over $\boldsymbol{x}_{1:T}$ with a bi-directional LSTM architecture (note this LSTM is used for inference not entropy coding), shown diagrammatically in Fig. 6. This allows $\boldsymbol{f}$ to depend on features from the entire segment. For the local state, the individual frame $\boldsymbol{x}_t$ is passed through the convolutional layers of $\boldsymbol{\mu_\phi}(\boldsymbol{x}_t)$ and a two-layer MLP infers $\boldsymbol{z}_t$ from the feature information of the individual frame. The decoder $\boldsymbol{\mu_\theta}(\boldsymbol{z}_t, \boldsymbol{f})$ first combines $(\boldsymbol{z}_t, \boldsymbol{f})$ with a multilayer perceptron (MLP) and then upsamples with a deconvolutional network. The prior models $p_{\boldsymbol{\theta}}(\boldsymbol{f})$ and $p_{\boldsymbol{\theta}}(\boldsymbol{z}_1)$ are parametrized by the density model defined in Appendix 6.1 of Ballé et al. (2018). The conditional prior $p_{\boldsymbol{\theta}}(\boldsymbol{z}_t \mid \boldsymbol{z}_{<t})$ in the LSTMP-LG architecture is modeled by a normal distribution which is convolved with uniform noise. The means and covariance of the normal distribution are predicted by an additional LSTM with hidden state $\boldsymbol{h}$.

Both encoders $\boldsymbol{\mu_\phi}(\cdot)$ have 5 convolutional (downsampling) layers. For layer $\ell = 1, 2, 3, 4$, the stride and padding are 2 and 1, respectively, and the convolutional kernel size is $4\times4$. The number of channels for layer $\ell = 1, 2, 3, 4$ are 192, 256, 512, 1024. Layer 5 has kernel size 4, stride 1, padding 0, and 3072 channels. The decoder architecture $\boldsymbol{\mu_\theta}$ is chosen to be asymmetric to the encoder with convolutional layers replaced with deconvolutional (upsampling) layers. For the Sprites toy video, the dimensions of $\boldsymbol{z}$, $\boldsymbol{f}$, and hidden state $\boldsymbol{h}$ are 64, 512 and 1024, respectively. For less sparse videos (BAIR and Kinetics600), the dimensions of $\boldsymbol{z}$, $\boldsymbol{f}$, and hidden state $\boldsymbol{h}$ are 256, 2048 and 3072, respectively.

**KFP-LG.** KFP-LG is also a proposed baseline model which incorporates both the global state $\boldsymbol{f}$ and local latent $\boldsymbol{z}_t$ but uses a weaker predictive model $p_{\boldsymbol{\theta}}(\boldsymbol{z}_t \mid \boldsymbol{z}_{t-1})$ for entropy coding. The main purpose of the KFP-LG model is to compare to the LSTMP-LG model which has a longer memory. The conditional prior $p_{\boldsymbol{\theta}}(\boldsymbol{z}_t \mid \boldsymbol{z}_{t-1})$ in KFP-LG is described by a deep Kalman Filter parametrized

by a three-layer MLP. The dimension at each layer of MLP is the same as the dimension of the latent state $z_t$. KFP-LG has the same encoder and decoder structures as the proposed LSTMP-LG model aforementioned. The only difference between KFP-LG and LSTMP-LG is that they employ different prior models for conditional entropy coding.

## C  LATENT VARIABLE DISTRIBUTION VISUALIZATION

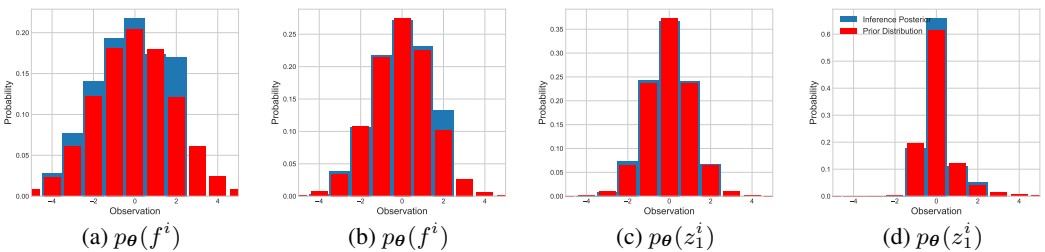

(a) $p_{\boldsymbol{\theta}}(f^i)$      (b) $p_{\boldsymbol{\theta}}(f^i)$      (c) $p_{\boldsymbol{\theta}}(z_1^i)$      (d) $p_{\boldsymbol{\theta}}(z_1^i)$

Figure 7: Empirical distributions of the posterior of inference model and ground truth prior model in one specific rate-distortion BAIR example.

In this appendix, we visualize the distribution of our prior model and compare to the empirical distribution of the posterior of the inference model estimated from data. In Fig. 7, we show the learned priors and the empirically observed posterior over two dimensions of the global latent state $\boldsymbol{f}$ and local latent state $\boldsymbol{z}$ in order to demonstrate that the prior is capturing the correct empirical distribution in low-bit rate of our model. From Fig. 7, we can see that the learned priors $p_{\boldsymbol{\theta}}(\boldsymbol{f})$ and $p_{\boldsymbol{\theta}}(\boldsymbol{z}_1)$ match the empirical data distributions well, which leads to low-bit rate encoding of the latent variables. As the conditional probability model $p_{\boldsymbol{\theta}}(\boldsymbol{z}_t \mid \boldsymbol{z}_{<t})$ is high dimensional, we do not display this distribution.

## D  ADDITIONAL PERFORMANCE EVALUATION

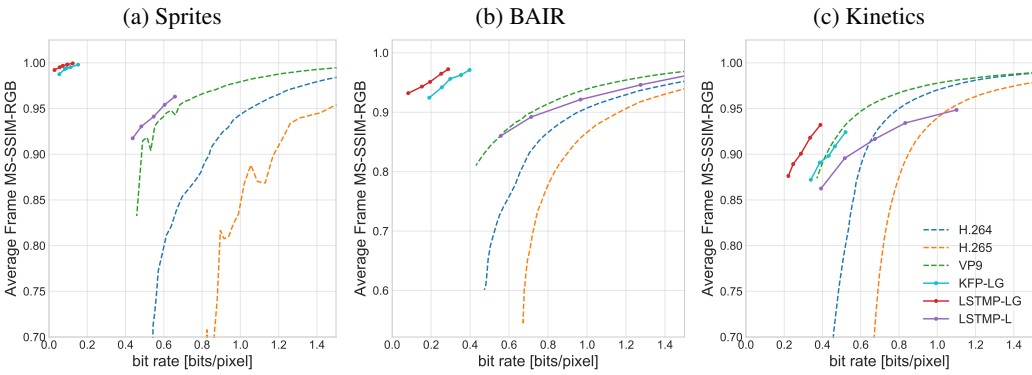

Figure 8: Rate-distortion curves on three datasets measured in MS-SSIM (higher corresponds to lower distortion). Legend shared. Solid lines correspond to our models, with LSTMP-LG proposed.

We also plot the average frame MS-SSIM with respect to the bit rate to quantitatively compare our models with traditional codecs. From Fig. 8, we can see that our LSTMP-LG method saves significantly more bits when trained on specialized content dataset (Sprites and BAIR) and achieves competitive result with respect to MS-SSIM when trained on general content dataset.

In Section 4, we trained our method on short video segments of $T = 10$ frames and evaluated classical codec performance on the same segments. For typical videos, somewhat longer segments tend to have less information per pixel than $T = 10$ segments, and standard video codecs are designed to take advantage of this fact. For this reason, we have presented codec performance, evaluated on

$T = 10$, 30, and 100 frame segments for the Kinetics data in Fig. 9. While codec performance improves for longer segments, we note that our method (trained and evaluated on 10 frames) is still comparable to modern codec performance evaluated on longer segments. Additionally, with proper design, our method could potentially be improved to achieve similar temporal performance scaling since longer segments usually have less information per pixel.

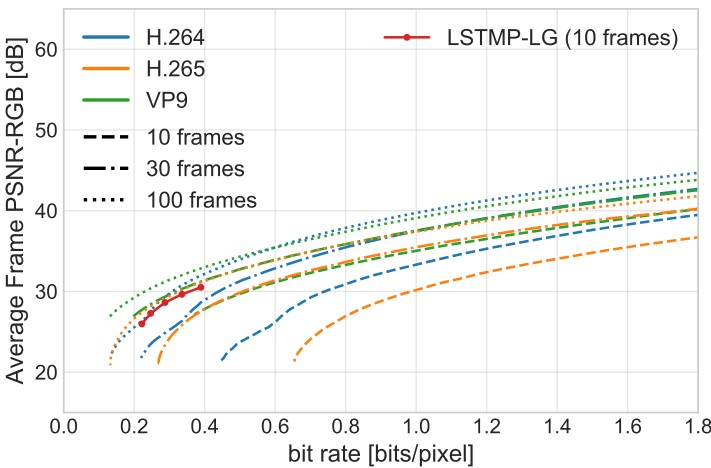

Figure 9: Rate-distortion curves on the Kinetics dataset measured in PSNR. Codec performance is evaluated on video segments of $T = 10$, 30, and 100 frames. Our best performing method (trained and evaluated on $T = 10$ frames) is shown in red for comparison.

