# OpenReview forum: "Deep Probabilistic Video Compression"
_ICLR.cc/2019/Conference_

### Official Review · AnonReviewer1 · 2018-11-01
**interesting, but very limited idea**

**Rating:** 6
**Confidence:** 5

**Review:**

This method deals with compressing tiny videos using an end-to-end learned approach. However, the paper has a significant number of limitations, which I will discuss below.

1. The method has only been trained on very small videos due to the fact that fully connected layers are used. I don't really understand why was this necessary, and it's not explained in the paper at all. Just this fact makes it completely infeasible for any "real" application.
2.  The evaluation was done on very limited domains. Of huge concern to me is the fact that very good results are presented on the sprites dataset. However, that dataset can be literally encoded by providing an index in a lookup table of sprites, so it's absolutely ludicrous to compare learned methods on that set to general video compression methods. The results look a lot less exciting when looking at the Kinetics 64x64 dataset.
3. The evaluation (again) is problematic because the results refer to PSNR. PSNR for video is a very overloaded term. In fact, just the way to compute PSNR is not very clear for video. Video compression papers in general compute it in one of two ways: take the mean squared error over all the pixels in the video, then compute PSNR; or compute per frame PSNR then average. Additionally, none of the papers in this domain use RGB, because the human visual system is much more sensitive to detail preservation (the Y/luminance channel) than they are to chroma (color) changes. When attempting to present results for video, I would recommend to use PSNR-Y (and explain which type it is!), while also mentioning which ITU recommendation is used for defining the Y channel (there are multiple recommendations).
4. It is not very clear how the global code is obtained. It is implied that all frames get processed in order to come up with f, but does this mean that they're processed via an LSTM model, or is there a single fully connected layer which takes as input all frames? In terms of modeling f, it sounds like the hyperprior model from Balle et al is employed, but again it's not clear to me how (is it modelling an entire video or a sequence?). I would really like to see a diagram for the network structure that computes f.

Ont he positives of the paper: I applaud the authors with respect to the fact that they made an effort to explain how the classical codecs were configured and being explicit about the chroma sampling that's employed.

I think all the problems I mentioned above can be fixed, so I don't want to reject the paper per se. If possible, should the authors address my concerns (i.e., add more details), I think this could be an interesting "toy" method.

---

> ### Author Response · Authors · 2018-11-15
> **Response to Reviewer 1 (1/2)**
>
> We would like to thank the reviewer for their time and feedback. Our response is detailed below.
>
> >1. The method has only been trained on very small videos due to the fact that fully connected layers are used. I don't really understand why was this necessary, and it's not explained in the paper at all. Just this fact makes it completely infeasible for any "real" application.
>
> We have openly pointed out that scalability is an open problem which we think can be solved with more research. We still believe there are some niche applications for video compression of small-scale content, e.g. thumbnail videos for previews.
>
> The core idea of our method is to probabilistically entropy code according to a deep generative model. The most successful generative models for videos in the current literature typically have fully connected components in the temporal prior (e.g. a fully connected recurrent network or lstm). Our proposed approach splits information into global and local latent states which allows for a very efficient compression in the small bitrate regime. By the usage of fully-connected networks, our method is able to capture non-local motion efficiently, which is hard to do solely with a local, convolutional model.
>
> >2.  The evaluation was done on very limited domains. Of huge concern to me is the fact that very good results are presented on the sprites dataset. However, that dataset can be literally encoded by providing an index in a lookup table of sprites, so it's absolutely ludicrous to compare learned methods on that set to general video compression methods. The results look a lot less exciting when looking at the Kinetics 64x64 dataset.
>
> We respectfully disagree, our method was evaluated on an unseen test set of Sprites videos, so the test videos could not be simply encoded as a lookup table.  We agree that the Sprites data set is a toy dataset since it has a low-dimensional description compared to a typical real-world video. However, there may be applications for which the video to be compressed lives on a lower-dimensional manifold, e.g. teleconferencing or sports videos. For such an application, it is beneficial for the video codec to learn such a manifold in order to more efficiently compress the video (we showed this for the low-dimensional but real-world BAIR data set). Since existing codecs are hand designed they do not possess this ability.
>
> >3. The evaluation (again) is problematic because the results refer to PSNR. PSNR for video is a very overloaded term. In fact, just the way to compute PSNR is not very clear for video. Video compression papers in general compute it in one of two ways: take the mean squared error over all the pixels in the video, then compute PSNR; or compute per frame PSNR then average. Additionally, none of the papers in this domain use RGB, because the human visual system is much more sensitive to detail preservation (the Y/luminance channel) than they are to chroma (color) changes. When attempting to present results for video, I would recommend to use PSNR-Y (and explain which type it is!), while also mentioning which ITU recommendation is used for defining the Y channel (there are multiple recommendations).
>
> We use the average per frame PSNR in RGB space in our work. Accordingly, our loss is phrased in RGB space and the video codecs are configured to operate in RGB mode (4:4:4 chroma sampling). We have clarified this point in the manuscript. While we acknowledge that it could be a good idea to use PSNR-Y as our performance metric, we have chosen to use PSNR-RGB to be consistent with the most closely-related papers from neural image compression (Balle et al. 2018, Minnen et al. 2018) which minimize RGB errors and report results in PSNR-RGB and neural video compression (Wu et al. 2018) (We contacted them and found that they also used average frame PSNR-RGB).

---

> > ### Author Response · Authors · 2018-11-15
> > **Response to Reviewer 1 (2/2)**
> >
> > >4. It is not very clear how the global code is obtained. It is implied that all frames get processed in order to come up with f, but does this mean that they're processed via an LSTM model, or is there a single fully connected layer which takes as input all frames? In terms of modeling f, it sounds like the hyperprior model from Balle et al is employed, but again it's not clear to me how (is it modelling an entire video or a sequence?). I would really like to see a diagram for the network structure that computes f.
> >
> > The architecture of the encoder is described in Appendix B. All frames in a segment are individually processed by a convolutional network and passed through an LSTM model which infers the global state f. A diagram of the network structure that computes f can be found in Appendix B of the revised version.
> >
> > We use a generic factorized probability density model from Balle et al. 2018 in order to model f (which, in their work, is used to entropy code the hyperprior latent variables). We do not use the hyperprior model because in our approach, the f latent variables are not spatially correlated. (The latent f’s do not capture the spatial structure of the video frames because the LSTM decorrelates the convolutional features.)

---

### Official Review · AnonReviewer2 · 2018-11-01
**Good approach to deep learning based video compression, but empirical section needs work**

**Rating:** 5
**Confidence:** 4

**Review:**

Summary
=======
This work on video compression extends the variational autoencoder of Balle et al. (2016; 2018) from images to videos. The latent space consists of a global part encoding information about the entire video, and a local part encoding information about each frame. Correspondingly, the encoder consists of two networks, one processing the entire video and one processing the video on a frame-by-frame basis. The prior over latents factorizes over these two parts, and an LSTM is used to model the coefficients of a sequence of frames. The compression performance of the model is evaluated on three datasets of 64x64 resolution: sprites, BAIR, and Kinetics600. The performance is compared to H.264, H.265, and VP9.

Review
======
Relevance (9/10):
-----------------
Compression using neural networks is an unsolved problem with potential for huge practical impact. While there has been a lot of research on deep image compression recently, video compression has not yet received much attention.

Novelty (6/10):
---------------
This approach is a straightforward extension of existing image compression techniques, but it is a reasonable step towards deep video compression.

What's missing from the paper is a discussion of how the proposed model would be applied to model video sequences longer than a few frames. In particular, the global latent state will be less and less useful as videos get longer. Should the video be split into multiple sequences treated separately? If yes, how should they be split and what is the impact on performance?

Empirical work (2/10):
----------------------
Unfortunately, the experiments focus too much on trying to make the algorithm look good at the expense of being less informative and potentially misleading.

Existing video codecs such as H.265 and software like ffmpeg are optimized for longer, high-resolution videos, but even the most realistic dataset used here (Kinetics600) only contains short (10 frames) low-resolution videos. I suggest the authors at least add the performance of classical codecs evaluated on the entire video sequence to their plots. The current reported performance can be viewed as splitting the videos into chunks of 64x64x10, which makes sense for an autoencoder which has been trained to learn a global representation of short videos, but is clearly not necessary and detrimental to the performance of the classical codecs. I think adding these graphs would provide a more realistic view of the current state of video compression using deep neural nets.

For the classical codecs, were the binary files stripped of any file format container and headers before counting bits? This would be crucial for a fair comparison, especially for small videos where the overhead might be significant.

More work could be done to ensure the reader that the hyperparameters of the classical codecs such as GOP or block size have been sufficiently tuned.

What is the frame rate of the videos used? I.e., how much time do 10 frames correspond to?

The videos were downsampled before cropping them to 64x64 pixels. What was the resolution before cropping?

The authors observe that the Kalman prior performs worse than the LSTM prior. This may be due to limitations of the encoder, which processes images frame-by-frame, which makes it hard to decorrelate frames while preserving information. I am wondering why the frame encoder is not at least processing one neighboring frame. (Note: A sufficiently powerful encoder could represent information in a fully factorial way; e.g. Chen & Gopinath, 2001).

Clarity:
--------
The paper is well written and clear.

---

> ### Author Response · Authors · 2018-11-15
> **Response to Reviewer 2 (1/2)**
>
> Thank you for your detailed response. We address each point individually below.
>
> >What's missing from the paper is a discussion of how the proposed model would be applied to model video sequences longer than a few frames. In particular, the global latent state will be less and less useful as videos get longer. Should the video be split into multiple sequences treated separately? If yes, how should they be split and what is the impact on performance?
>
> Correct, the video could be divided up into segments, with global states specific to these segments. Especially for longer sequences, choosing optimal segments - i.e. pieces of the video that are well described by a single global state - is an important problem to solve for future work towards a practical deep learning based video codec.
>
> >Unfortunately, the experiments focus too much on trying to make the algorithm look good at the expense of being less informative and potentially misleading.
>
> >Existing video codecs such as H.265 and software like ffmpeg are optimized for longer, high-resolution videos, but even the most realistic dataset used here (Kinetics600) only contains short (10 frames) low-resolution videos. I suggest the authors at least add the performance of classical codecs evaluated on the entire video sequence to their plots. The current reported performance can be viewed as splitting the videos into chunks of 64x64x10, which makes sense for an autoencoder which has been trained to learn a global representation of short videos, but is clearly not necessary and detrimental to the performance of the classical codecs. I think adding these graphs would provide a more realistic view of the current state of video compression using deep neural nets.
>
> Our approach is based on preprocessing a longer video by dividing it into segments of length T. Every segment has a unique global state, and T local states. In our paper, we used T=10 frames per segment for computational convenience and memory limitations of our available hardware. This choice is comparable to the sequence length (GOP size) chosen in (Wu et al., 2018).
>
> The optimal segment length depends on the input data. When comparing our approach with classical codecs, we used the same short video segments across all methods. While using 10 frames per segment may be short, we expect both classical codecs and neural network approaches to benefit in similar ways from longer segments.
>
> To address the concerns about potentially misrepresenting codec performance, we have added the codec performance curves for different segment lengths to Fig. 8 in Appendix D. For typical footage, we found that the performance of classical codecs increases with longer sequences and saturates before 100 frames. We note that our method, trained and evaluated on T = 10 frames, remains comparable to H.264/H.265 tested on T = 100 frames even though T=100 segments typically contain less information per pixel than T=10 segments. VP9 outperforms our method when tested on T=100 frames.
>
> >For the classical codecs, were the binary files stripped of any file format container and headers before counting bits? This would be crucial for a fair comparison, especially for small videos where the overhead might be significant.
>
> We were unable to find a way to easily determine the size of the header information. If the reviewer could share the command to find out such information with us, we would be very glad to analyze this aspect. To address the concerns about header size, we have added codec performance curves for longer sequences (T=100) frames to Fig. 8 in Appendix D where header information is expected to be a much smaller fraction of the file size.
>
> >More work could be done to ensure the reader that the hyperparameters of the classical codecs such as GOP or block size have been sufficiently tuned.
>
> We used the video codec in the standard way, without excessive parameter tuning. Our main objective was to show that deep learning architectures, designed from scratch, can be comparable to standard codecs in certain regimes.
>
> >What is the frame rate of the videos used? I.e., how much time do 10 frames correspond to?
>
> The Sprites dataset does not have a physical time since it is generated video. The Kinetics dataset contains YouTube videos with variable frame rates, ranging between 24-60 fps.
>
> >The videos were downsampled before cropping them to 64x64 pixels. What was the resolution before cropping?
>
> The videos were cropped to 1:1 aspect ratio and then downsampled to 64x64. The original videos were various resolutions ranging from 426 x 240 to 1920 x 1080.

---

> > ### Author Response · Authors · 2018-11-15
> > **Response to Reviewer 2 (2/2)**
> >
> > >The authors observe that the Kalman prior performs worse than the LSTM prior. This may be due to limitations of the encoder, which processes images frame-by-frame, which makes it hard to decorrelate frames while preserving information. I am wondering why the frame encoder is not at least processing one neighboring frame. (Note: A sufficiently powerful encoder could represent information in a fully factorial way; e.g. Chen & Gopinath, 2001).
> >
> > The full encoder, with both local and global state, is processing an entire segment of video, which includes neighboring frames. Moreover, it is not necessary for the per-frame latent variables to be completely decorrelated temporally because the temporal correlation is taken into account by the learned temporal prior (which conditions on the previous frame(s)). The temporal redundancy is removed from the bit stream by entropy coding the per-frame latents according to the learned temporal prior distribution.
> >
> > Reference:
> > Chao-Yuan Wu, Nayan Singhal, and Philipp Krahenbuhl.  Video compression through image interpolation. European Conference on Computer Vision, 2018.

---

### Official Review · AnonReviewer3 · 2018-11-02
**An interesting proposal for deep-learning-based video compression, but somewhat limited experimental results and unclear applicability.**

**Rating:** 6
**Confidence:** 5

**Review:**

The paper is well written and the basic ideas are reasonably well explained and supported. However, several aspects are insufficiently explained. Several examples follow.

In Figure 1, it is not clear at all how the bitstream is formed; frames 1 to T are compressed jointly with frame t; but frame t is part of the set of frames from 1 to T. How the global state updated when compressing frame t+1? Using frames 2 to T+1?

Writing that you use a Laplacian distribution because l1 regularized loss typically outperforms the l`2 loss for autoencoding images is clearly an insufficient justification, if not backed by experiments or references. Moreover, the authors seem to confuse regularization with loss; by using a Laplace density for the generative model, they are using a l1 loss, not an l1 regularizer.

There is absolutely no information about implementation details.

The video sequences used in the experiments are extremely small, both in spatial and temporal terms. A collection of 10 64*64 frames has fewer pixels than even a moderately sized still image. As the authors acknowledge, standard video codecs are far from optimized for video sequences of this size, making the comparisons unfair. The extreme compression results on the sprites and BAIR datasets may be quite misleading, since the data lives in a very low dimensional manifold, due to the simplicity of the scenes. For the more realistic Kinetics dataset, the proposed method is competitive with H264 and H265, but only in a very limited range of bit rates. In fact, the authors do not explain why they have not show results for wider ranges of bitrates.

---

> ### Author Response · Authors · 2018-11-15
> **Response to Reviewer 3**
>
> We would like to thank the reviewer for their time and feedback. Below is our response to each point made by the review.
>
> >In Figure 1, it is not clear at all how the bitstream is formed; frames 1 to T are compressed jointly with frame t; but frame t is part of the set of frames from 1 to T. How the global state updated when compressing frame t+1? Using frames 2 to T+1?
>
> We assume that the video gets divided up into segments of T frames. Every segment has exactly one global state - which is jointly computed and not updated incrementally - and T local states. When forming the bitstream, the global state is formed by running a bi-directional LSTM over all T frames while the local states are formed on a per frame basis. When decoding, each frame can be reconstructed using the global state and its specific local state. We will add a corresponding explanation to the revised version.
>
> >Writing that you use a Laplacian distribution because l1 regularized loss typically outperforms the l2 loss for autoencoding images is clearly an insufficient justification, if not backed by experiments or references. Moreover, the authors seem to confuse regularization with loss; by using a Laplace density for the generative model, they are using a l1 loss, not an l1 regularizer.
>
> In our experiments, we found that an L1 loss tended to produce sharper image reconstructions than an L2 loss. We refer to references (Isola et al., 2016;  Zhao et al., 2015) which suggest the benefits of using the L1 loss for image reconstruction. We acknowledge that the Laplacian corresponds to using an L1 loss, not an L1 regularizer, and this statement was a typo in the manuscript which will be corrected. We have also added these references to the manuscript in order to justify the use of the L1 loss.
>
> >There is absolutely no information about implementation details.
>
> We have added information about implementation details in Appendix B of the revised version.
>
> >The video sequences used in the experiments are extremely small, both in spatial and temporal terms. A collection of 10 64*64 frames has fewer pixels than even a moderately sized still image. As the authors acknowledge, standard video codecs are far from optimized for video sequences of this size, making the comparisons unfair. The extreme compression results on the sprites and BAIR datasets may be quite misleading, since the data lives in a very low dimensional manifold, due to the simplicity of the scenes. For the more realistic Kinetics dataset, the proposed method is competitive with H264 and H265, but only in a very limited range of bit rates. In fact, the authors do not explain why they have not show results for wider ranges of bitrates.
>
> We acknowledge the fact that practical video compression for high resolution content is an extremely complex problem that requires solving many sub problems. In this work, which arguably is the first one to use deep probabilistic modeling for end-to-end video compression, we have used small video data in order to focus on exploring new ways to model temporal redundancy.
>
> Standard codecs are indeed not optimized for such small videos. However, standard video codecs drastically outperform other baselines, such as neural image compression or JPEG compression per frame, on such data. In lack of other baselines on this type of data, we provide standard codec results in order to demonstrate that our method is efficiently capturing temporal correlations. In our conclusions, we already acknowledged the fact (but will highlight this even more) that more work needs to be done in order to outperform standard codecs on all resolutions.
>
> The reviewer correctly points out that certain data (such as Sprites and BAIR) may live on a lower-dimensional manifold as compared to general content video. We emphasize that this may also be true for other types of data such as video conferencing or sports broadcasting, so it may be beneficial for a specialized-content codec to learn such a lower-dimensional manifold in a data-driven approach.
>
> Regarding the range of bitrates, the reason for the limited range of bitrates on the kinetic dataset is due to limited GPU memory. We have made this point more clear in the limitations section (before the conclusions) of the revised manuscript. The highest quality setting is limited by the size of the latent space. General content video requires a larger latent dimension than specialized video, and the latent space dimension could not be increased any further due to such hardware limitations. Resolving this problem is an interesting and important avenue of further research.
>
> Isola, P., Zhu, J. Y., Zhou, T., & Efros, A. A. (2016). Image-to-image translation with conditional adversarial networks. arXiv preprint arXiv:1611.07004.
>
> Zhao, Hang, et al. "Loss functions for image restoration with neural networks." IEEE Transactions on Computational Imaging3.1 (2017): 47-57.

---

### Meta-Review · Area_Chair1 · 2018-12-14

**Confidence:** 4
**Recommendation:** Reject

**Metareview:**

The proposed method is compressing video sequences with an end-to-end approach, by extending a variational approach from images to videos. The problem setting is interesting and somewhat novel. The main limitation, as exposed by the reviewers, is that evaluation was done on very limited and small domains. It is not at all clear that this method scales well to non-toy domains or that it is possible in fact to get good results with an extension of this method beyond small-scale content. There were some concerns about unfair comparisons to classical codes that were optimized for longer sequences (and I share those concerns, though they are somewhat alleviated in the rebuttal).

While the paper presents an interesting line of work, the reviewers did present a number of issues that make it hard to recommend it for acceptance. However, as R1 points out, most of the problems are fixable and I would advise the authors to  take the suggested improvements (especially anything related to modeling longer sequences) and once they are incorporated this will be a much stronger submission.